



# Improving sub-seasonal forecast skill of meteorological drought: a weather pattern approach

Doug Richardson[1,2], Hayley J. Fowler[2], Chris G. Kilsby[2], Robert Neal[3], Rutger Dankers[3,4]

[1]CSIRO Oceans & Atmosphere, Hobart, Australia, 7001
[2]School of Engineering, Newcastle University, Newcastle-upon-Tyne, NE1 7RU, United Kingdom
[3]Weather Science, Met Office, Exeter, EX1 3PB, United Kingdom
[4]Environmental Research, Wageningen University & Research, Wageningen, 6708 PB, Netherlands

*Correspondence to*: Doug Richardson (doug.richardson@csiro.au)

**Abstract.** Dynamical model skill in forecasting extratropical precipitation is limited beyond the medium-range (around 15
days), but such models are often more skilful at predicting atmospheric variables. We explore the potential benefits of using weather pattern (WP) predictions as an intermediary step in forecasting UK precipitation and meteorological drought on sub-seasonal time scales. MSLP forecasts from the ECMWF ensemble prediction system (ECMWF-EPS) are post-processed into probabilistic WP predictions. Then we derive precipitation estimates and dichotomous drought event probabilities by sampling from the conditional distributions of precipitation given the WPs. We compare this model to the direct precipitation and
drought forecasts from ECMWF-EPS and to a baseline Markov chain WP method. A perfect-prognosis model is also tested to illustrate the potential of WPs in forecasting. Using a range of skill diagnostics, we find that for 31- and 46-day lead-times, dynamical, and to a lesser extent Markov, model forecasts using WPs can achieve higher skill scores that the non-WP method, particularly for precipitation. Forecast skill scores are generally modest (rarely above 0.4), although those for the perfect-prognosis model highlight the potential predictability of precipitation and drought using WPs, with certain situations yielding
skill scores of almost 0.8, and drought event hit and false alarm rates of 70% and 30%, respectively.

## 1 Introduction

Droughts are a recurrent climatic feature in the UK. Severe events, such as those in 1975-76, 1995 and 2010-12, had significant implications for many sectors, including agriculture, water resources and the economy, as well as for ecosystems and natural habitats (Marsh, 1995; Marsh *et al.*, 2007; Rodda and Marsh, 2011; Kendon *et al.*, 2013). To mitigate the effects of drought,
it is crucial that relevant sectors plan ahead, and drought forecasts have an important role in designing these strategies. Despite this, there is very little published research on UK drought prediction, and studies have predominantly focussed on hydrological drought (Wedgbrow *et al.*, 2002; Wedgbrow *et al.*, 2005; Hannaford *et al.*, 2011).

Meteorological drought is challenging to predict using dynamical ensemble prediction systems (Yoon *et al.*, 2012; Dutra *et al.*, 2013; Yuan and Wood, 2013; Mwangi *et al.*, 2014; Lavaysse *et al.*, 2015). This is primarily due to the complex processes
involved in precipitation formation, making it a difficult variable to forecast beyond short lead-times (Golding, 2000; Cuo *et al.*, 2011; Smith *et al.*, 2012; Saha *et al.*, 2014). At longer lead-times, dynamical model skill in predicting atmospheric variables tends to be much higher (Saha *et al.*, 2014; Scaife *et al.*, 2014; Vitart, 2014; Baker *et al.*, 2018). This has led researchers to investigate the potential of using atmospheric forecasts as a precursor to predicting precipitation-related hazards (Lavers *et al.*, 2014; Lavers *et al.*, 2016; Baker *et al.*, 2018).

Weather pattern (WP; also called weather types, circulation patterns and circulation types) classifications are a candidate for such an application. A WP classification consists of a number of individual WPs, which are typically defined by an atmospheric variable and represent the broad-scale atmospheric circulation over a given domain (Huth *et al.*, 2008). They can be used to make general predictions of local-scale variables such as wind speed, temperature and precipitation and are a tool for reducing atmospheric variability to a few discrete states. WP classifications have mainly been studied in the context of extreme hydro-
meteorological events (Hay *et al.*, 1991; Wilby, 1998; Bárdossy and Filiz, 2005; Richardson *et al.*, 2018a; Richardson *et al.*,


2018b), and as a tool for analysing historical and future changes in atmospheric circulation patterns (Hay *et al.*, 1992; Wilby, 1994; Brigode *et al.*, 2018). See Huth *et al.* (2008) for a comprehensive review of WP classifications.

Until recently, the capability of dynamical models to predict WP occurrences had been little researched. Ferranti *et al.* (2015) evaluated the forecast skill of the medium-range European Centre for Medium-Range Weather Forecasts ensemble prediction

system (ECMWF-EPS) (Buizza *et al.*, 2007; Vitart *et al.*, 2008) using WPs. They objectively defined four WPs according to daily 500 hPa geopotential heights over the North Atlantic – European sector. Model forecasts of this variable for October through April between 2007 and 2012 were then assigned to the closest matching WP using the root-mean-square difference. Verification scores indicated that there was superior skill for predictions initialised during negative phases of the North Atlantic Oscillation (NAO) (Walker and Bliss, 1932). Similarly, WPs were used to evaluate the skill of the Antarctic Mesoscale

Prediction System by Nigro *et al.* (2011).

To support weather forecasting in the UK in the medium- to long range, the Met Office use a WP classification, MO30, in a post-processing system named "Decider" (Neal *et al.*, 2016). Using a range of ensemble prediction systems, forecast mean sea-level pressure (MSLP) fields over Europe and the North Atlantic Ocean are assigned to the best-matching WP according to the sum-of-squared differences between the forecast MSLP anomaly and WP MSLP anomaly fields. Decider therefore

produces a probabilistic prediction of WP occurrences for each day in the forecast lead-time. Decider has various operational applications: predicting the possibility of flow transporting volcanic ash originating in Iceland into UK airspace, highlighting potential periods of coastal flood risk around the British Isles (Neal *et al.*, 2018) and as an early-forecast system for fluvial flooding (Richardson *et al.*, in review).

For Japan, Vuillaume and Herath (2017) defined a set of WPs according to MSLP. These WPs were used to refine bias-

correction procedures, via regression modelling, of precipitation from two global ensemble forecast systems. The authors found that improvements from the bias-correction method using WPs was strongly dependent on the WP, but overall superior to the global (non-WP) method. Relevant to this study, Lavaysse *et al.* (2018) predicted monthly drought in Europe using a WP-based method. They aggregated ECMWF-EPS daily reforecasts of WPs to predict monthly frequency anomalies of each WP. For each 1° grid cell, the predictor was chosen to be the WP that corresponded to the maximum absolute temporal

correlation between the monthly WP frequency of occurrence anomaly and the monthly Standardised Precipitation Index (SPI) (McKee *et al.*, 1993). Using this relationship, the model predicted drought in a grid cell when 40% of the ECMWF-EPS ensemble members forecast a Standardised Precipitation Index (SPI; McKee *et al.*, 1993) value below -1. Compared to direct ECMWF-EPS drought forecasts, the WP-based model was more skilful in north-eastern Europe during winter, but less skilful for central and eastern Europe during spring and summer. Over the UK, the WP model appeared to be superior for north-

western regions in winter, but inferior in summer, although scores for the latter were of low magnitude.

The aforementioned studies have all considered daily WPs. An example of WPs defined on the seasonal time-scale was presented by Baker *et al.* (2018). The authors analysed reforecasts of UK regional winter precipitation between the winters of 1992-93 and 2011-12 using GloSea5, which has little raw skill in forecasting this variable (MacLachlan *et al.*, 2015). GloSea5 has, however, been shown to skilfully forecast the winter NAO (Scaife *et al.*, 2014). Baker *et al.* (2018) exploited this by

constructing two winter MSLP indices over Europe and the North Atlantic, and reforecasts of these indices were derived from the raw MSLP fields. A simple regression model then related these indices to regional precipitation and produced more skilful forecasts than the raw model output.

In this study, we shall explore the potential for utilising a WP classification (specifically MO30) in UK meteorological drought prediction. We shall predict WPs using two models, ECMWF-EPS and a Markov chain, from which precipitation and drought

forecasts will be derived. These models will be compared to direct precipitation and drought forecasts from ECMWF-EPS. We also run an idealised, perfect prognosis model that uses WP observations rather than forecasts as an 'upper benchmark' to



assess the upper limit of the usefulness of the WP classification. Section 2 contains details of the data sets used, including describing the creation of a WP reforecast data set. Section 3 describes the models in detail and the forecast verification procedure. In Sect. 4, we shall present the results and in Sect. 5, we draw some conclusions and make recommendations for

future work.

## 2 Data

We use a Met Office WP classification called MO30 (Neal *et al.*, 2016). WPs in MO30 were defined by clustering 154 years (1850-2003) of daily MSLP anomaly fields into 30 distinct states. The data were extracted from the European and North Atlantic daily to multidecadal climate variability (EMULATE) data set (Ansell *et al.*, 2006) in the domain 30° W-20° E; 35°-

70° N, with a spatial resolution of 5° latitude and longitude. These 30 WPs are therefore representative of the 30 most common patterns of daily atmospheric circulation over Europe and the North Atlantic (Fig. 1), and they are ordered such that WP1 is the most frequently occurring WP annually, while WP30 is the least frequent. A consequence of the clustering process and ordering is that the lower-numbered WPs have lower-magnitude MSLP anomalies and are more common in the summer than in the winter, and vice versa for the higher-numbered WPs (Richardson *et al.*, 2018a)(Neal *et al.*, 2016).

For this analysis, we have created a 20-year daily WP probabilistic reforecast data set. We use the sub-seasonal to seasonal (S2S) project (Vitart *et al.*, 2017) data archive, which, through ECMWF, hosts reforecast data for a multitude of variables and by a range of models from around the globe. In particular, we use ECMWF-EPS, which is a coupled atmosphere-ocean-sea-ice model with a lead-time of 46 days. The horizontal atmospheric resolution is roughly 16 km up to day 15 and 32 km beyond this. The model is run at 00Z, twice weekly (Mondays and Thursdays) and has 11 ensemble members for the reforecasts

(compared to 51 members for the real-time forecasts). For further details, refer to the model webpage (ECMWF, 2017). We use daily reforecasts of MSLP between 02 January 1997 and 28 December 2016, inclusive, with the same domain and resolution as MO30. These fields are converted to anomalies by removing a smoothed climatology and subsequently assigned to the closest matching MO30 WP via minimising the sum-of-squared differences. Both the MSLP climatology and the WP definitions are the same as those used by Neal *et al.* (2016) to ensure consistency. We compare this against an observed WP

time series to measure forecast skill. For this, WPs are assigned from 00Z SLP fields from the ERA-Interim reanalysis data set (Dee *et al.*, 2011) to align with the ECMWF-EPS forecast times.

As observed precipitation, we use the Met Office Hadley Centre UK Precipitation (HadUKP) data set (Alexander and Jones, 2000). For nine regions covering the UK (Fig. 2), we use daily precipitation series from 1979 to 2017. We discretise the data into precipitation intervals ("bins") defined in Table 1; see the supporting material for further information. The large region

sizes in HadUKP are suitable both for analyses of drought, which is typically considered a regional rather than localised event, and for MO30 because they correspond to the large-scale circulation patterns that the WPs represent. From the S2S archive, we extract ECMWF-EPS precipitation reforecasts for the same dates as the WP reforecast data set. The data have a resolution of 0.5° latitude and longitude; grid cells are assigned to whichever of the nine HadUKP regions the cell centres lie in and by taking the daily mean of all cells over each region, we produce a probabilistic reforecast data set of precipitation for each of

the HadUKP regions. These data are discretised in the same way as the HadUKP data.

## 3 Methods

### 3.1 Weather pattern forecast models and verification procedure

For WP forecasts, we compare two models. The first is ECMWF-EPS, which we shall refer to as EPS-WP (in practice this is the WP reforecast data set discussed in the previous subsection). The second model is a 1000-member, first-order,

nonhomogeneous Markov chain, with separate transition matrices for each month. This is similar to the Markov model used



for a simulation study by Richardson *et al.* (2018b), who found it was able to reasonably replicate the observed frequencies of occurrences of the MO30 WPs. Full details of the Markov model are given in the supporting material.

To evaluate WP forecast skill we use the Jensen-Shannon divergence (JSD), suitable for measuring the distance between two probability distributions (Lin, 1991). It is based on information entropy, which is used to measure uncertainty. An information-theoretic approach to verification is not widespread, although there is some published research on the topic (Leung and North, 1990; Kleeman, 2002; Roulston and Smith, 2002; Ahrens and Walser, 2008; Weijs *et al.*, 2010; Weijs and Giesen, 2011). The JSD will be used to measure the forecast performance by quantifying the distance between distributions of the observed and forecast WP frequencies. The JSD is based on the Kullback-Leibler divergence (KLD) (Kullback and Leibler, 1951). Let $P$ and $Q$ be two discrete probability distributions. The KLD from $Q$ to $P$ is given by:

$$D_{KL}(P||Q) = -\sum_{i=1}^{I} P_i \log_2 \frac{Q_i}{P_i},$$

Equation 1

measured in bits (i.e. a binary unit of information). In our application $I = 30$, the number of WPs and $P = (p_{f,1}, \dots, p_{f,30})$ and $Q = (q_{f,1}, \dots, q_{f,30})$ are the vectors of observed and forecast WP relative frequencies, respectively. (Because these are relative frequencies, $\sum P = 1$ and $\sum Q = 1$.) As there would inevitably be some cases where the model predicts no occurrences of some WPs (i.e. when $Q$ contains zeros), $D_{KL}(P||Q)$ will be undefined at times. Using the JSD avoids this problem; it is defined as:

$$D_{JSD}(P||Q) = \frac{1}{2} D_{KL}(P||M) + \frac{1}{2} D_{KL}(Q||M),$$

Equation 2

where $M = (P + Q)/2$. Unlike the KLD, the JSD is symmetric i.e. $D_{JSD}(P||Q) \equiv D_{JSD}(Q||P)$. Also, $0 \leq D_{JSD}(P||Q) \leq 1$, with a score of zero indicating $P$ and $Q$ are the same (a perfect forecast). Equation 2 gives the JSD for a single forecast-event pair; to obtain the average JSD for all forecasts we take the mean of all forecast-event pairs. Skill is evaluated separately for each month, with the middle date of each forecast period used to assign the month. We calculate forecast skill for lead-times of 16, 31 and 46 days. We use the JSD to compare WP forecast skill of EPS-WP and the Markov model, considering each lead-time separately.

### 3.2 Precipitation and drought forecast models

We compare four models (Table 2), three of which are forecast models, while one model is a perfect prognosis model. All models are considered at the same lead-times as the WP predictions. Two of the forecast models are driven first by a WP component: EPS-WP and the Markov model described above. The perfect prognosis model, Perfect-WP, is used as an 'upper benchmark' with (future) observed WPs as input, rather than forecast WPs. It is an idealised model that cannot be used operationally, but it allows us to assess the potential usefulness of WPs in precipitation and drought forecasting. Precipitation is estimated from the WP predictions (or observations in the case of Perfect-WP) by sampling from the conditional distributions of precipitation given each WP. As we discretised the precipitation, these conditional distributions reflect the observed relative frequencies of each precipitation interval occurring. The sampling procedure is done for each ensemble member and each day in the forecast lead-time, with the results summed across all members and days to provide probabilistic forecasts of summed precipitation intervals (Table 1). A full description of this method is detailed in the supporting information. The fourth model (the third forecast model) is the direct ECMWF-EPS precipitation forecasts (EPS-P), processed to provide probabilistic predictions of regional precipitation intervals as described earlier.



### 3.3 Precipitation forecast verification

To evaluate precipitation forecast performance we use the ranked probability score (RPS) (Epstein, 1969; Murphy, 1971). We
express the RPS as the ranked probability skill score (RPSS) using

$$RPSS = 1 - \frac{RPS}{RPS_{ref}},$$

Equation 3

Where $RPS_{ref}$ is the score of a climatological forecast, which in our case is the climatological event category (i.e. precipitation interval) relative frequencies (PC). A perfect score is achieved when $RPSS = 1$, which is also the upper limit. Negative
(positive) values indicate the forecast is performing worse (better) than $RPS_{ref}$.

### 3.4 Drought forecast verification

We evaluate model performance in predicting dichotomous drought/non-drought events. We define two classes of drought severity. The first class, mild drought, is when precipitation sums (over the length of the considered lead-time) are below the 30.9th percentile of the summed precipitation distribution. The second class is moderate drought, with such sums being below
the 15.9th percentile. These percentiles are calculated for each region and month using the whole data set from 1979 through 2017, and are chosen as they correspond to SPI values of -0.5 and -1, respectively.

#### 3.4.1 The Brier Skill Score

We use three verification techniques to assess skill in predicting droughts. The first is the Brier Skill Score (BSS). The BSS is based on the Brier Score (BS) (Brier, 1950), which measures the mean-square error of probability forecasts for a dichotomous
event, in this case the occurrence or non-occurrence of drought. The BS is converted to a relative measure, or skill score, by setting

$$BSS = 1 - \frac{BS}{BS_{ref}},$$

Equation 4

where $BS_{ref}$ is the score of a reference forecast given by the quantiles associated with each drought threshold, 0.309 for mild
drought and 0.159 for moderate drought. As with the RPSS, a perfect score is achieved when $BSS = 1$ and negative (positive) values indicate the forecast is performing worse (better) than $BS_{ref}$.

#### 3.4.2 Reliability diagrams – forecast reliability, resolution and sharpness

The BS can be decomposed into reliability, resolution and uncertainty terms (Murphy, 1973):

$$BS = reliability - resolution + uncertainty,$$

Equation 5

enabling a more in-depth assessment of forecast model performance. Reliability diagrams offer a convenient way of visualising the first two of these terms (Wilks, 2011). These diagrams consist of two parts, which together show the full joint distribution of forecasts and observations. The first element is the calibration function, $g(o_1|p_i)$, for $i = 1, \dots n$, where $o_1$ indicates the event (here, a drought) occurring and the $p_i$ are the forecast probabilities. The calibration function is visualised by plotting the
event relative frequencies against the forecast probabilities and indicates how well calibrated the forecasts are. We split the forecast probabilities into 10 bins (subsamples) of 10% probability and the mean of all forecast probabilities in each bin is the


value plotted on the diagrams (Bröcker and Smith, 2007). Points along the 1:1 line represent a well-calibrated, *reliable*, forecast, as event probabilities are equal to the forecast probabilities and suggest that we can interpret our forecasts at 'face value'. If the points are to the right (left) of the diagonal, the model is over-forecasting (under-forecasting) the number of drought events.


The forecast *resolution* can also be deduced from the calibration function. For a forecast with poor resolution, the event relative frequencies $g(o_1|p_i)$ only weakly depend on the forecast probabilities. This is reflected by a smaller difference between the calibration function and the horizontal line of the climatological event frequencies and suggests that the forecast is unable to resolve when a drought is more or less likely to occur than the climatological probability. Good resolution, on the other hand,

means that the forecasts are able to distinguish different subsets of forecast occasions for which the subsequent event outcomes are different to each other.

The second element of reliability diagrams is the refinement distribution, $g(p_i)$. This expresses how confident the forecast models are by counting the number of times a forecast is issued in each probability bin. This feature is also called *sharpness*. A low-sharpness model would overwhelmingly predict drought at the climatological frequency, while a high-sharpness model

would forecast drought at extreme high and low probabilities, reflecting its level of certainty with which a drought will or will not occur, independent of whether a drought actually does subsequently occur or not.

### 3.4.3 Relative operating characteristics

As a final diagnostic we use the relative operating characteristic (ROC) curve (Mason, 1982; Wilks, 2011), which visualises a model's ability to discriminate between events and non-events. Conditioned on the observations, the ROC curve may be

considered a measure of potential usefulness – it essentially asks what the forecast is, given that a drought has occurred. The ROC curve plots the hit rate (when the model forecasts a drought and a drought subsequently occurs) against the false alarm rate (when the model forecasts a drought but a drought does not then occur). We compute the hit rate and false alarm rate for cumulative probabilities between 0% and 100% at intervals of 10%. A skilful forecast model will have a hit rate greater than a false alarm rate, and the ROC curve would therefore bow towards the top-left corner of the plot. The ROC curve of a forecast

system with no skill would lie along the diagonal, as the hit rate and false alarm rate would be equal, meaning the forecast is no better than a random guess. The area under the ROC curve (AUC) is a useful scalar summary. AUC ranges between zero and one, with higher scores indicating greater skill.

### 4 Results

To reduce information overload, we do not show results for every combination of region, lead-time and drought class. Key

results not shown will be conveyed via the text. We aggregate the precipitation results from monthly to three-month seasons for visual clarity and combine regional results for the ROC and reliability diagrams for the same reason.

### 4.1 WP forecasts

We find that EPS-WP is more skilful at predicting the WPs than the Markov model for every month and every lead-time, although the difference in skill between the two models decreases as the lead-time increases. The skill difference between

models is much larger for a lead-time of 16 days compared to a lead-time of 46 days. For a 46-day lead-time, the difference in skill is negligible for May through October; in fact, these months have the smallest differences in JSD for all lead-times. This is presumably because the summer months are associated with fewer WPs compared to winter (Richardson *et al.*, 2018a), resulting in a more skilful Markov model due to higher transition probabilities.

An interesting result is how JSD scores for both models, especially for Markov, decrease as the lead-time increases (Fig. 3),

suggesting an improvement in skill with lead-time. This is the opposite of the expected (and usual) effect and is probably





because both the observations and the forecasts tend towards climatology at longer lead-times. The JSD measures the distance between probability distributions, and at the shorter lead-times, the forecast relative frequency distribution tends to be much noisier compared to the observed relative frequency distribution i.e. a greater number of different WPs are predicted than observed. As the lead-time is increased, the observations become noisier and as a result the JSD tends to score the differences

between these distributions as more similar (a smaller divergence).

### 4.2 Precipitation forecasts

During summer and spring, all three forecast models are well matched, although for a 16-day lead-time Markov is the least skilful. For this lead-time, EPS-P mostly scores similarly to EPS-WP, although it has higher skill for some regions (Figs. 4b and 4c) and even outperforms Perfect-WP for several regions in summer (Fig. 4c). At lead-times of 31 and 46 days, there is

little difference in forecast model skill during spring and summer, although in summer NI and SWE appear to benefit from dynamical WP predictions (i.e. EPS-WP), as do the four eastern regions from any kind of WP forecast (EPS-WP and Markov; Fig. 5). On the other hand, using WP predictions is to the detriment of precipitation forecast skill in spring for SEE, as shown by the superior performance of EPS-P (Fig. 5). This split between the east and west is also found by Lavaysse *et al.* (2015), who used ECMWF-EPS to predict meteorological drought with a one month lead-time.

For winter and autumn, EPS-WP is the most skilful forecast model except when considering a 16-day lead-time, for which EPS-P is often the best performer. Scotland benefits most from the use of EPS-WP, as even at the shortest lead-time this model is superior (Figs. 4a and 4d). Note that the skill of the WP forecasts matter, as Markov is associated with poor precipitation skill at this lead-time, which corresponds to its low skill in forecasting the WPs compared to EPS-WP (Fig. 3). EPS-WP is the most skilful model for 31- and 46-day lead-times; EPS-P and Markov score fairly evenly overall for a 31-day lead-time, with

the former model the least skilful for a 46-day lead-time (Fig. 5). The difference in skill between EPS-WP and Markov is much larger for northern and western regions, particularly in winter. Therefore the improvement in skill by predicting the WPs with a dynamical model, rather than Markov (Fig. 3), translates to a spatially non-uniform gain in skill for precipitation, with western and northern regions the principal beneficiaries. However, it is difficult to say why this is the case, as from the JSD scores alone we do not know whether EPS-WP is better at predicting all WPs, or just some of them. Similarly, as the RPSS is

for all forecast dates, we can't be sure whether the improvement in precipitation forecast skill comes from an improvement over all periods or if the gain is made for predictions of dry or wet periods; drought forecast analysis results in the next subsection will go some way to answering this.

Unsurprisingly, Perfect-WP is uniformly the most skilful precipitation 'forecast' model for all regions, seasons and lead-times, except for some regions and seasons with a 16-day lead-time. At this shortest lead-time, Perfect-WP is the most skilful in all

cases during winter (Fig. 4a) and in all cases except NS during spring (Fig. 4b) and NEE during autumn (Fig. 4d), for which EPS-P is the most skilful. The only season in which Perfect-WP does not have the most skill for most regions is during summer, when EPS-P is superior (Fig. 4c). For lead-times of 31 and 46 days, perfectly predicting WPs would enable by far the most skilful precipitation estimates of any model, for all regions and seasons (Fig. 5). This model is obviously not practical, but the results serve to show that WPs are a potentially useful tool in medium-range precipitation forecasting.

The key conclusions from this subsection are that, for winter and autumn, precipitation forecasts are notably more skilful when derived from dynamical predictions of WPs compared to either simple statistical WP predictions or direct precipitation forecasts from a dynamical model. Furthermore, the relative gain in skill is greater for longer lead-times, mainly as a result of a notable drop in skill for EPS-P when comparing a 31-day to a 46-day lead-time, whereas other forecast models' score changes are less severe. For spring and summer, EPS-P is marginally the most skilful model at a 16-day lead-time, with little to choose

between all three forecast models at the longer lead-times. A potential reason for the lower skill of WP-based models compared to EPS-P in summer is that the WPs associated with this season tend to be less clear-cut in terms of being associated with dry





or wet conditions (Richardson *et al.*, 2018a), possibly as a result of their higher intra-WP variability compared to winter WPs (Richardson *et al.*, 2018b). Only WP6, WP8 and WP9 are distinctly dry or wet and so precipitation estimates from summer WPs may not be appropriate for periods of non-normal precipitation.

### 4.3 Drought forecasts

#### 4.3.1 Forecast accuracy

Forecast accuracy for mild and moderate drought is qualitatively similar to those of general precipitation in terms of regional and lead-time differences. EPS-WP is overall the most skilful model, although this is less the case for a 16-day lead-time, for the three regions in East England and for most regions in spring and summer. Only the results for predicting moderate drought at the 46-day lead-time are presented (Fig. 6). This is because the results for mild drought are more similar to the RPSS results than those of moderate drought, those for the 16-day lead-time are the least useful for drought prediction (which tends to be focussed on longer-range forecasts) and those for the 31-day lead-time are qualitatively similar to the 46-day lead-time.

For the shortest lead-time, EPS-P has the highest accuracy for predicting winter and autumn drought of both classes, except in Scotland, for which EPS-WP has the highest skill. Indeed, EPS-WP has the highest skill for the other lead-times during these seasons (Fig. 6). However, a key difference is that eastern England droughts are at least as accurately predicted by EPS-P as by EPS-WP for the two longer lead-times (Fig. 6), whereas for precipitation forecasting the latter tend to be more accurate (Fig. 5). Difference in model skill is lower for spring and summer drought forecasts, particularly for moderate drought (Fig. 6). In fact, for this drought class, there is very little or no gain in skill by using WPs at 31- and 46-day lead-times for spring and summer compared to EPS-P (Fig. 6). Furthermore, at these lead-times both models are less skilful than issuing climatological drought probabilities (shown by their negative BSS), except for spring predictions of eastern and southern droughts. This suggests that, during spring and summer, deriving precipitation from predicted WPs may be useful if forecasting mild drought, but not for more severe droughts.

#### 4.3.2 Relative operating characteristics

All models are better able to discriminate between drought and non-drought events than random chance, with Perfect-WP the most able and Markov the least able, subject to similar caveats regarding lead-time and season as for the BSS and RPSS results. During summer and spring, EPS-P has the highest AUC of any of the three forecast models (Figs. 7 and 8), and for a 16-day lead-time scores similarly to Perfect-WP (not shown). On the other hand, EPS-WP has the highest skill during winter and autumn at the other lead-times, particularly for mild drought. Markov is consistently the least suitable model for predicting drought according to the ROC curve, although still represents a better method of doing so than random chance.

A use of the ROC curve is to provide end-users with information on how to apply the considered forecast models. As the plotted points on each curve indicate the hit rate and false alarm rate associated with predicting droughts at each probability interval, they can be used to make an informed decision in selecting a probability threshold for issuing a drought forecast. For example, should a forecaster choose to issue a mild drought warning in winter at a 20% probability level and 46-day lead-time (Fig. 7), then they would expect EPS-WP to achieve a hit rate roughly double that of the false alarm rate (60% and 30%, respectively). EPS-P, meanwhile, shows a slightly higher hit rate but at the expense of a higher false alarm rate (65% and 40%). The idealised benchmark model (Perfect-WP) achieves an outstanding score – roughly a 75% hit rate compared to a 10% false alarm rate. For mild drought, a 20% probability threshold for EPS-WP and EPS-P achieves at least 60% hit rates at all lead-times, whereas for moderate drought, this threshold will only achieve such rates at a 16-day lead-time and during autumn for all lead-times. In general, it appears that these low probability thresholds yield the best compromise between hits and false alarms, although in practice, the costs (e.g. financial) associated with false alarms and missed events will determine how responders use these probabilities.



### 4.3.3 Forecast reliability, resolution and sharpness

EPS-WP is the most reliable forecast model, and while all three WP-driven forecast models tend to under-forecast droughts, EPS-P only does so for lower probability thresholds, with the higher thresholds resulting in this model over-forecasting. This is particularly true for shorter lead-times and during winter, although is still clear for 31-day lead-times in some seasons (Figs. 9 and 10). Sometimes EPS-WP follows the same pattern as EPS-P and over-forecasts drought occurrence for higher predicted probabilities (e.g. Figs. 9c, e, g and 10c). However, the total number of forecasts issued in these intervals is generally smaller than for EPS-P, as the refinement distributions show most clearly for mild drought (Fig. 9). This means the corresponding points of the calibration function are less reliable for EPS-WP (and Markov) due to smaller sample sizes (Bröcker and Smith, 2007). In fact, all three WP-based models have occasions when there are no issued forecasts with certain probabilities. These are high probabilities for Perfect-WP and EPS-WP (Figs. 10c and e) but can be as low as between 30% and 40% for Markov (Figs. 10e and g). As such, although EPS-WP appears the most reliable model from looking only at the calibration function, there is less certainty of this fact for moderate drought and for higher forecast probabilities. This erratic behaviour of the conditional event relative frequencies is most obvious in Fig. 10c and is explained by the very low sample sizes of forecasts issued with anything but a small probability (Fig. 10e) (Wilks, 1995). An interesting result is that forecasts from EPS-WP are more reliable than from Perfect-WP (Figs. 9 and 10), despite having lower accuracy (e.g. Fig. 6). As a more skilful BSS is composed of smaller reliability and larger resolution terms (Eq. 5), it follows that the resolution of Perfect-WP is sufficiently large to overcome the larger reliability term compared to EPS-WP and yield an overall more accurate forecast model. These under- or over-forecasting biases must be taken into account by an operational forecaster using these models.

A key difference apparent from the calibration function relates to the ability of the models to identify subsets of forecast situations where the subsequent event relative frequencies are different, i.e. the forecast resolution. An almost completely consistent feature across all lead-times and drought classes is the poorer resolution of EPS-P, particularly obvious in autumn (Figs. 9g and 10g), with the conditional event relative frequencies quite clearly closer to the climatological average compared to the other models. This should be considered in conjunction with the sharpness of the forecast, which is relatively high for this model as shown by the numbers of issued extreme probabilities, particularly those in the upper-tail (Figs. 9h and 10h). This combination of poor resolution and high sharpness indicates "overconfidence" (Wilks, 2011) – on the occasions that EPS-P issues a forecast indicating the likelihood of a drought is very high, the actual likelihood of a drought subsequently occurring is lower. To compensate for this overconfidence, a user would adjust the probabilities to be less extreme to make the forecasts more reliable.

We can compare these refinement distributions to those of the Markov model, which exhibits low sharpness, overwhelmingly predicting droughts at the climatological frequency (second column of Figs. 9 and 10). This means that the Markov model is not a useful operational tool in these situations, as similar forecasts could be obtained simply by using the climatological drought frequency. The refinement distributions for EPS-WP show that for mild drought in winter and spring and for moderate drought in all seasons, the model predicts droughts with low probabilities the majority of the time (Figs. 9b, d and 10b, d, f, h). For mild drought in summer and autumn, however, this model mostly issues forecasts close to the climatological frequency, although not nearly as regularly as the Markov model (Fig. 9f, h). As with adjusting for bias, a forecaster can use model resolution and sharpness when assessing drought forecast probabilities output by a model.

### 5 Discussion and conclusions

We have compared the performance of a dynamical forecast system (EPS-WP) and a first-order Markov model in predicting WP occurrences over a range of lead-times, showing that the dynamical model is always more skilful, although the difference in skill reduces with lead-time. From these WP predictions, we derived precipitation forecasts and compared them to direct precipitation predictions from the dynamical system (EPS-P). EPS-P has the highest overall skill in precipitation and drought





forecasts for a 16-day lead-time, whereas EPS-WP predictions provided the greatest skill for longer 31- and 46-day lead-times. We also demonstrated the potential in improving WP forecasts further by showing that an idealised, perfect prognosis model

(Perfect-WP) would provide much more skilful precipitation and drought forecasts, with high hit rates and low false alarm rates.

From assessing reliability diagrams we found that WP-based models only issue binary drought forecasts with either very low probabilities or probabilities close to the climatological average. In particular, there is little to gain in using the Markov model in mild drought prediction over the climatological frequency, as it tends to issue drought forecasts with this probability anyway.

EPS-P has the highest sharpness, predicting drought occurrence with a wide range of probabilities. In particular, it issues greater numbers of high-probability drought forecasts compared to WP-based methods. However, this model also has poor resolution, indicating it is an overconfident forecast model. Overall, drought forecasts issued by EPS-WP are the most reliable, i.e. the forecast probabilities are most similar to the subsequent event probabilities (they "mean what they say") (Wilks, 2011). Perfect-WP tends to under-forecast the number of drought events, while EPS-P over-forecasts drought events, particularly for

moderate drought. These reliability diagrams are therefore useful to aid users in adjusting for an over- or under-forecasting bias.

Given the results presented here, we would recommend the use of EPS-WP for the following drought forecast situations.

- Winter and autumn 31- and 46-day forecasts.
- Winter and autumn 16-day forecasts for Scotland (ES, NS and SS).
- Spring and summer 16-day forecasts for ES.
- Summer 31- and 46-day forecasts of mild drought for eastern and southern regions.

EPS-P is recommended for:

- Winter and autumn 16-day forecasts for all regions except those in Scotland.
- Spring and summer 16-day forecasts for all regions except ES
- Spring 31- and 46-day forecasts for all regions except those in Scotland.

Otherwise, the use of climatological drought frequencies represents the most parsimonious (in terms of skill versus model complexity) choice for:

- Summer 31- and 46-day forecasts for mild drought in northern and western regions and moderate drought in all regions.
- Spring 31- and 46-day forecasts for Scotland.

Focussing on the 31- and 46-day lead-times (that are more useful for drought prediction than 16-day forecasts), winter and autumn are clear-cut, with EPS-WP recommended for every region. Summer is more complex. Mild droughts are best predicted by EPS-WP for the eastern and southern regions, but drought climatological frequencies are suggested over the forecast models for western and northern regions and more severe droughts for all regions. In spring, climatology is also

recommended for Scotland, with the use of EPS-P for the remaining regions.

The higher skill of EPS-WP during winter (and possibly autumn) is probably due to the typically higher skill that medium- to long-range dynamical forecast systems have in predicting atmospheric variables in this season compared to other seasons (Scaife *et al.*, 2014; MacLachlan *et al.*, 2015; Neal *et al.*, 2016). In fact, by forecasting a set of eight WPs derived from MO30, Neal *et al.* (2016) found that ECMWF-EPS exhibited greater skill in winter than summer. Furthermore, the relationship

between the NAO (which is the primary mode of North Atlantic/European atmospheric circulation) and precipitation is stronger in this season (Hurrell and Deser, 2009; Lavers *et al.*, 2010; Svensson *et al.*, 2015). This is particularly true for western

regions (Jones *et al.*, 2013; Svensson *et al.*, 2015; van Oldenborgh *et al.*, 2015; Hall and Hanna, 2018), which potentially explains the greater difference in precipitation and drought forecast skill between EPS-WP and EPS-P in these seasons. The skill of precipitation forecasting using observed WPs (Perfect-WP) is also lower for eastern regions than western regions in

winter, implying that MO30 is not as suited for representing precipitation in the east. Perhaps this is because the WPs are more closely related to the NAO in this season, compared to other teleconnection patterns. As Hall and Hanna (2018) showed, the NAO is not the only important teleconnection pattern influencing UK precipitation. However, in general forecast skill is lower for eastern regions independent of the model.

By analysing the skill of an idealised 'forecast' model that assumes perfect WP predictions, we have demonstrated the potential
for using WP forecasts to derive precipitation and drought predictions. Currently, dynamical models such as the ECMWF system used here represent the best method of predicting WPs. Moreover, the ECMWF reforecast data used here had 11 ensemble members, whereas the live forecasts are run with 51 members. Therefore, an operationalised version of the models might improve forecast skill or better represent uncertainty. A useful piece of further research would be to assess the forecast skill of other models, and even multi-model ensembles, at predicting MO30 WPs or other WP classification systems. Another

potential method to improve precipitation and drought forecast skill would be to alter the process by which precipitation is derived from the WPs. Here we have sampled from the entire conditional distribution of precipitation given the WP and season, but this may not be the optimal way of estimation. It is possible that other factors influence the precipitation from WPs, such as slowly-varying atmospheric and oceanic processes. For example, it would be interesting to see if conditioning the distributions further on the state of the NAO index, or some North Atlantic SST index, and sampling precipitation from these,

would improve forecast skill. This is potentially most useful in predicting more severe forms of drought (D2 in this study), for which skill from current models is lower than for mild drought.

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



| Daily precipitation | | 30-day precipitation sums | |
|---|---|---|---|
| $p_b$ | Range of precipitation, $x$ (mm) | $s_c$ | Range of summed precipitation, $y$ (mm) |
| $p_1$ | 0 | $s_1$ | $0 \leq y < 10$ |
| $p_2$ | $0 < x \leq 1$ | $s_2$ | $10 < y \leq 20$ |
| ... | Intervals of 1mm | ... | Intervals of 10mm |
| $p_{11}$ | $9 < x \leq 10$ | $s_{25}$ | $240 < y \leq 250$ |
| $p_{12}$ | $10 < x \leq 15$ | $s_{26}$ | $250 < y \leq 300$ |
| $p_{13}$ | $15 < x \leq 20$ | ... | Intervals of 50mm |
| $p_{14}$ | $20 < x \leq 30$ | | |
| ... | Intervals of 10mm | | |

Table 1: Range of daily precipitation, $x$, for each bin $p_b$ and of 30-day precipitations sums, $y$, for each bin $s_c$.

| Model | WP component | Precipitation component |
|---|---|---|
| Markov | Predicted using a first-order Markov chain | Estimated by sampling from conditional distributions of precipitation given the WPs. |
| EPS-WP | Predicted by assignment of forecast SLP fields from ECMWF-EPS | |
| Perfect-WP | Observed WPs | |
| EPS-P | - | Forecast by ECMWF-EPS |

Table 2: Details of the four models. Markov, EPS-WP and EPS-P are forecast models and Perfect-WP is a perfect prognosis model that cannot be used for forecasting.

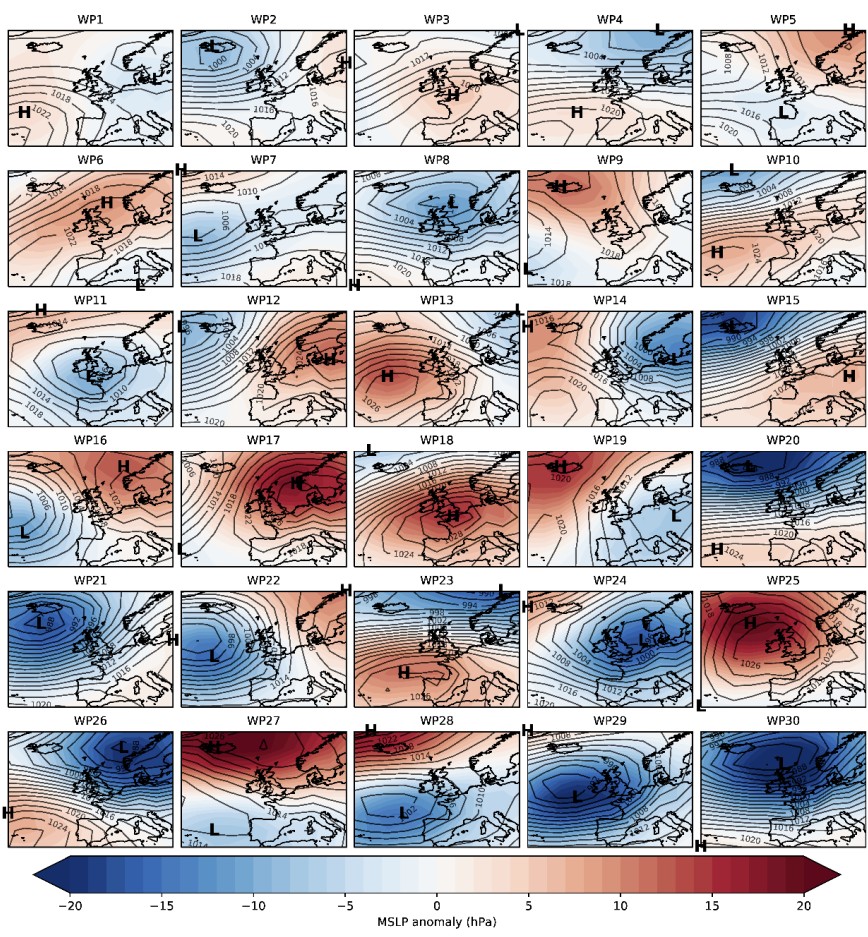

Figure 1: Weather pattern (WP) definitions according to mean sea-level pressure (MSLP) anomalies (hPa). The black contours are isobars showing the absolute MSLP values associated with each weather pattern, with the centres of high and low pressure also indicated. From Richardson *et al.* (2018b).




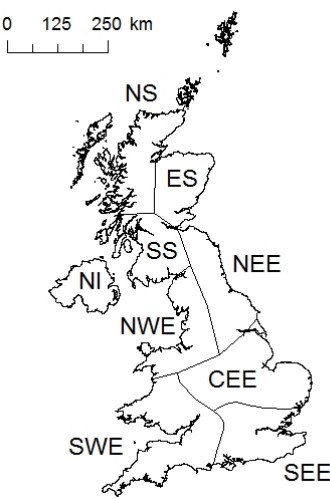

Figure 2: HadUKP regions: northeast England (NEE), central and east England (CEE), southeast England (SEE), southwest England and southern Wales (SWE), northwest England and northern Wales (NWE), Northern Ireland (NI), southwest Scotland (SS), northern Scotland (NS) and eastern Scotland (ES).

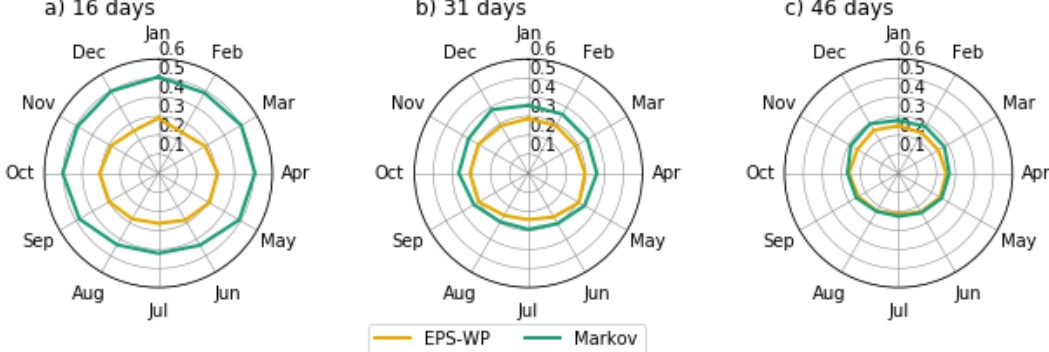

Figure 3: Jensen-Shannon Divergence scores for EPS-WP and Markov models for three lead-times.


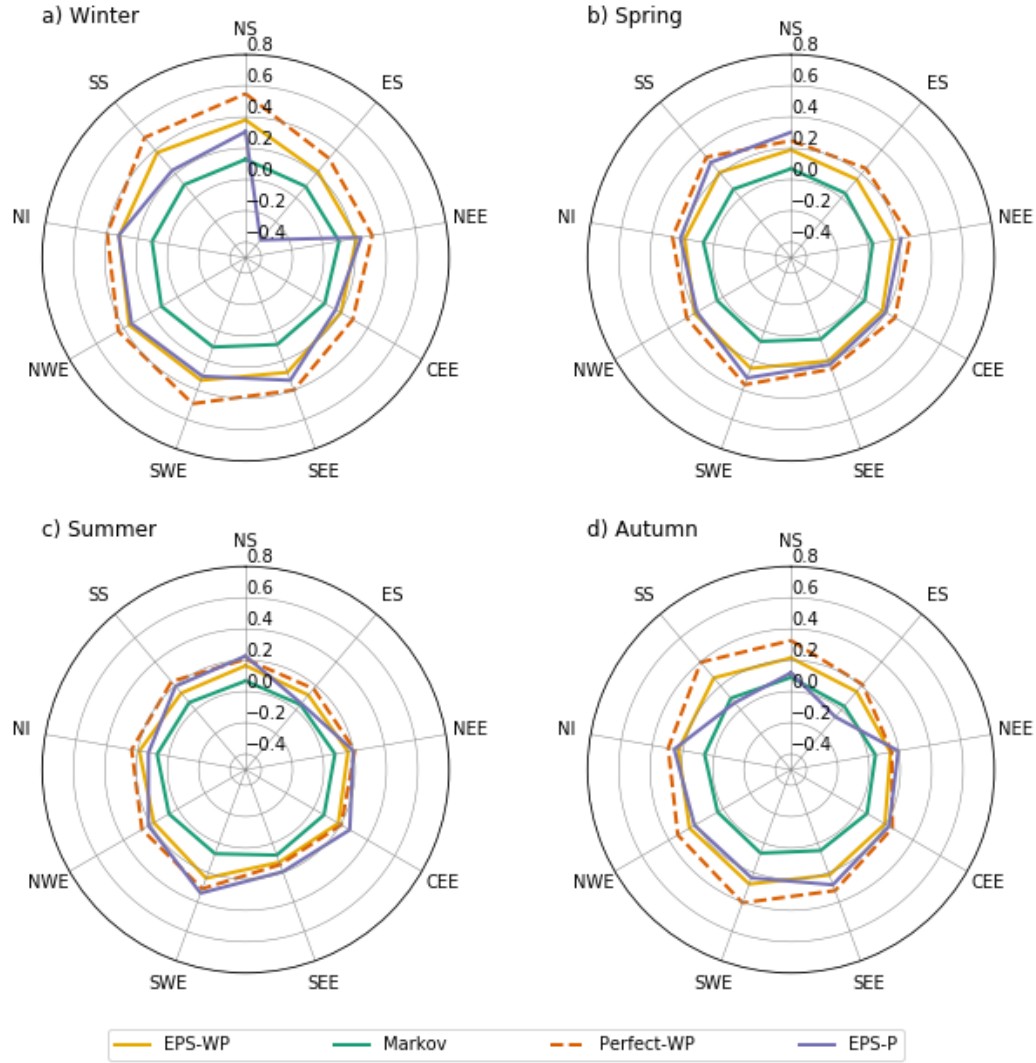

Figure 4: Ranked Probability Skill Scores by region and season for the three forecast models (EPS-WP, Markov and EPS-P) and the idealised model (Perfect-WP). Lead-time is 16 days. Scores lower than -0.5 are omitted for visual clarity. The omitted scores are for EPS-P in ES during spring (-0.54).


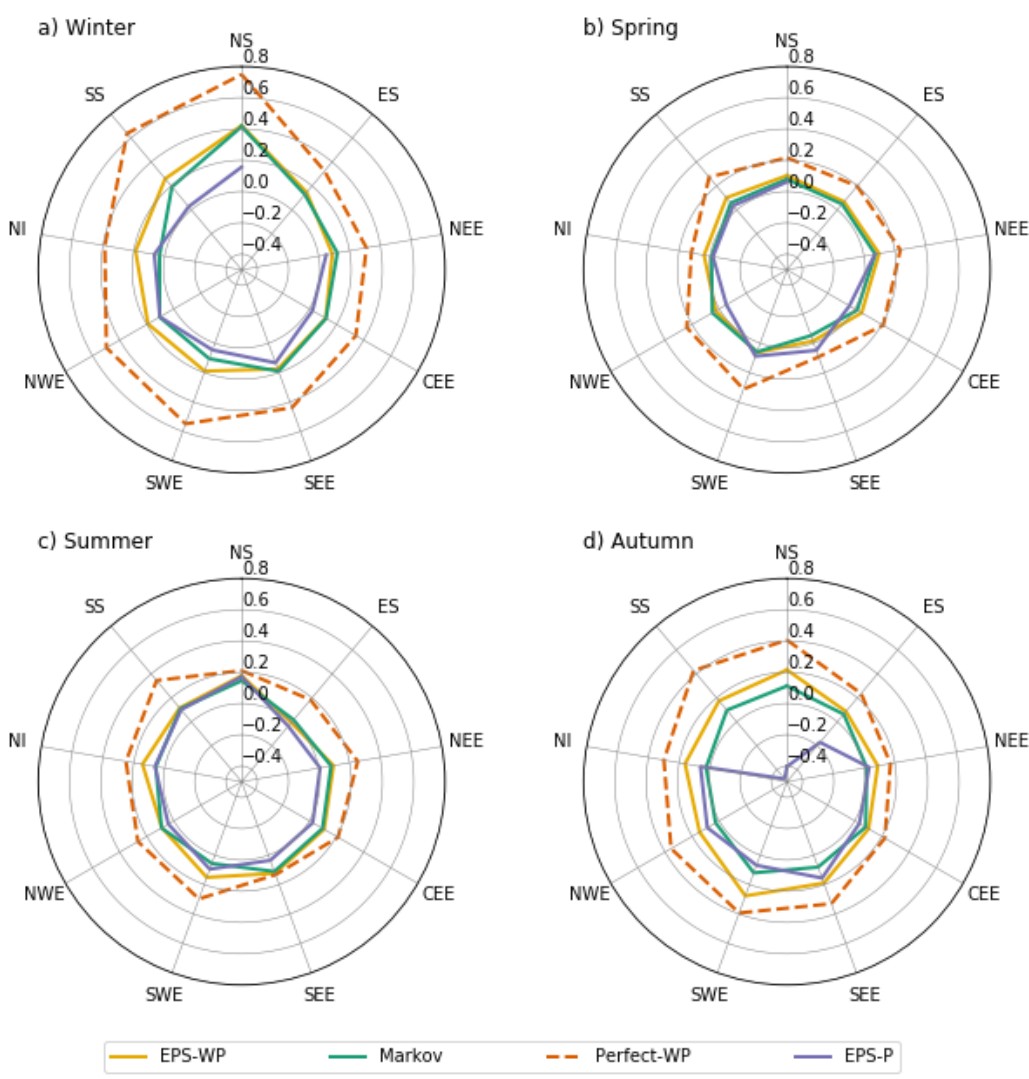

Figure 5: As Fig. 4 but for a lead-time of 46 days. The omitted scores are for EPS-P in ES during winter (-1.15) and spring (-1.37).


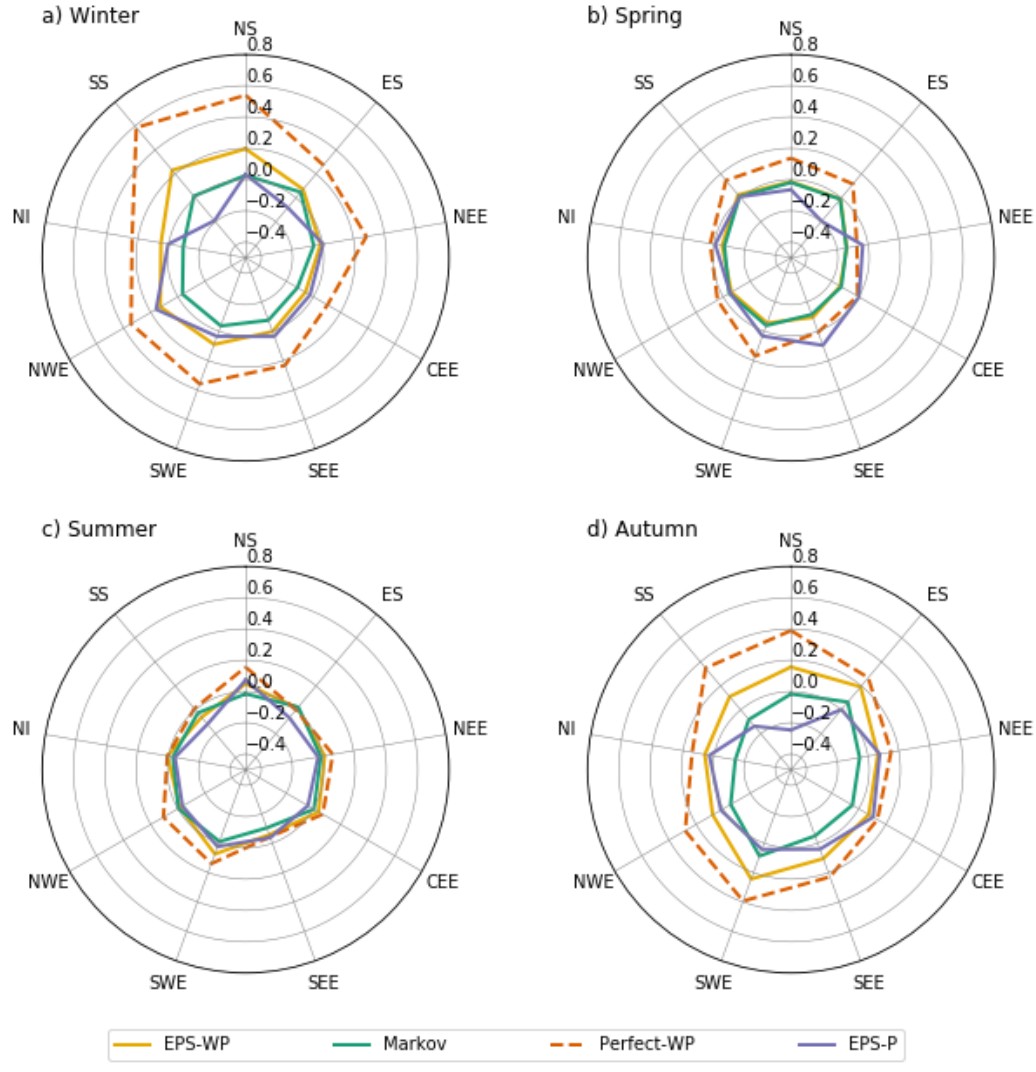

Figure 6: Brier Skill Score (BSS) by region and season for moderate drought for the three forecast models (EPS-WP, Markov and EPS-P) and the idealised model (Perfect-WP). Lead-time is 46 days.



Figure 7: Relative operating characteristics (ROC) curves and area under ROC curve (AUC) for mild drought with a 46-day lead-time. Annotated values indicate drought forecast probability thresholds.

Figure 8: As Fig. 7 but for moderate drought.

Figure 9: Calibration functions (first column) and refinement distributions (second column) for mild drought with a 31-day lead-time. For the calibration function diagrams, the solid diagonal line indicates perfect reliability and the dashed horizontal line the event relative frequency for mild drought (0.309).

Figure 10: As Fig. 9 but for moderate drought (event relative frequency of 0.159).