# Peer review of "Improving sub-seasonal forecast skill of meteorological drought: a weather pattern approach"

_Natural Hazards and Earth System Sciences, 2019_

## Referee Comment (RC1) · Christophe Lavaysse (Referee) · 2 Aug 2019

Review of the study "Improving sub-seasonal forecast skill of meteorological drought: a weather pattern approach" by Richardson et al. This study aims at analysing the predictability of meteorological droughts over UK and the potential interest of using predictors based on weather patterns. The authors conclude about the improvement of the forecasts by using this approach, and depending the seasons and regions, they provide recommendations to forecasters. This study is well documented, the figures and the text are clear and the statistics are robusts. After a careful reading, I recommend to publish this study that bring innovative and interesting results after substantial revisions. I provide here my recommandations:

[Figure]

Major comments: To clarify the different forecasts, I suggest to add a schema of the different forecast systems. The assignment procedure (from the forecasted WP to precipitation) is not clear enough and part of it should be moved from the Sup. Mat. to the main document.

Figures 4, 5 and 6: Since the scores are depending the regions, I would suggest to plot maps (with one value per region) instead of radar-plots. That will also provide more accurate information about the spatial variability of the scores.

It is quite surprising to use the same WPs for all the year long since there is a strong seasonal cycle. What are the results when splitting the year in 2 or 4 seasons? The use of several classification could improve the predictions, for instance in Spring and Summer (L289, Fig. 6b and c, Fig. 7b and c).

Detailed comments: l102: it is not clear if there is a post-processing of the reforecasts. Do you observe any drift with lead time ? Is there a bias between the distribution of assigned WP for short and long lead time ?

l111: same question for the forecasted precipitation. Is there a correction/post-processing applied?

4.1 I think the improvement of forecast with lead time deserves more attention. This result should be better analyzed. The classification of WP is done with reanalysis, correct? The sentences, "the observation and the forecasts tend towards climatology at longer lead time' and 'As the lead-time is increased, the observations become noisier ...' sound weirds to me. There is no 'lead time' for the observations. Please clarify.

L238 How do the authors explain the fact that EPS-WP outperforms Perfect-WP in summer? That could reflect a bias in the forecasted WP compared to the observed ones.

L250 "The difference in skill . . . for northern and western regions..." this will be more visible if the authors use maps instead of radar-plots.

L265 Maybe the sentences are a bit too optimistic. Indeed, for several regions, the differences between EPS-WP and EPS-P do not look significant (ES, NEE, SEE in winter for different lead times). The potential benefit of the use of WP (perfect-WP) could be also discussed since if the WP are not well forecasted and if the processes between WP and precipitation is well represented in the model, that means the main mistake of the model is the same when using forecasted WP and precipitation. That could limit the interest of using such predictors. This should be discussed.

L266 Does "simple statistical WP prediction" mean Markov here? I suggest to keep the same name of the experience.

L279 and Fig 6: Please remind to the readers that mild droughts mean here. Also I am a bit confused about the definition of droughts with lead time d-46. Droughts are calculated with 30-d cumulated periods. Since the authors used the Extended ensemble, how they calculated droughts with 16 and 46-d lead time? Please clarify.

L284 (and elsewhere): Because of the radar-plots it is quite complicated to locate the regions (here Scotland). That requires constantly back and forth to the UK map (Fig. 2).

L292 It is not clear why the authors conclude "... but not for more severe droughts." since these results are not shown nor discussed previously. Does it mean the same results with severe droughts provide negative BSS? These results could be discussed and added in Sup Mat.

L313 "EPS-WP is the most reliable forecast model" The authors should clarified that perfect-WP is the most reliable but not a real forecast.

L325 "... forecasts from EPS-WP are more reliable than from Perfect-WP..." According to several figures, I am not convinced with that conclusions (Fig. 9c, 9g, 10a, 10c, 10g)

Discussion section: In the recommendation section, the authors should redefine their drought definition. Also these conclusions could be different if they change the definition of WP (by splitting the year in 2 or 4 seasons).

---

## Referee Comment (RC2) · Anonymous Referee #2 · 26 Sep 2019

[referee-annotated manuscript omitted]

---

## Author Comment (AC1) · 23 Oct 2019

**Response to RC1**

*Review of the study "Improving sub-seasonal forecast skill of meteorological drought: a weather pattern approach" by Richardson et al. This study aims at analysing the predictability of meteorological droughts over UK and the potential interest of using predictors based on weather patterns. The authors conclude about the improvement of the forecasts by using this approach, and depending the seasons and regions, they provide recommendations to forecasters. This study is well documented, the figures and the text are clear and the statistics are robusts. After a careful reading, I recommend to publish this study that bring innovative and interesting results after substantial*

[Figure]

*revisions.*

**Response:** We thank the referee for their detailed review of our manuscript and hope to satisfactorily address their comments and concerns, as detailed below. Please note there is a supplement to this document.

**Major comments**

**Comment:** *To clarify the different forecasts, I suggest to add a schema of the different forecast systems. The assignment procedure (from the forecasted WP to precipitation) is not clear enough and part of it should be moved from the Sup. Mat. to the main document.*

**Response:** We agree that adding a schematic of the different forecast systems and expanding on the WP-to-precipitation detail in the main document would be useful to the reader.

**Changes to manuscript:** We have created a schematic for the four precipitation/drought forecast systems (Fig 2 in the supplement). We tested graphical schematics, but the number of graphics needed (many of which look similar – MSLP anomaly fields, for example) resulted in them being difficult to interpret. Therefore, we created a textual schematic. Table 2 is no longer required so we have removed it.

We have removed the assignment procedure from the Sup. Mat. To Section 3.2 of the manuscript.

**Comment:** *Figures 4, 5 and 6: Since the scores are depending the regions, I would suggest to plot maps (with one value per region) instead of radar-plots. That will also provide more accurate information about the spatial variability of the scores.*

**Response:** We chose the radar plots to reduce the number of sub-plots per figure. However, we accept the arguments from both referees that maps would be easier to interpret for a variety of reasons, not least for those with less familiarity of the UK regions used in our study.

**Changes to manuscript:** We have changed these figures to maps (see Figs. 4, 5, 6, 7, S1 and S2 in the supplement). This has enabled us to label the regions on each plot. Fig. 2 is therefore redundant and we have removed it.

**Comment:** *It is quite surprising to use the same WPs for all the year long since there is a strong seasonal cycle. What are the results when splitting the year in 2 or 4 seasons? The use of several classification could improve the predictions, for instance in Spring and Summer (L289, Fig. 6b and c, Fig. 7b and c).*

**Response:** There are several choices that must be made when deriving WP classifications, such as the domain, the spatial and temporal resolution of the data, the number of WPs etc., each of which is a trade-off. Whether to derive separate classifications for each season is also a choice. Classifying on each season might yield WPs that are more representative of the changes in MSLP that occur seasonally, but at the expense of reducing the sample size (which can be critical for machine learning methods such as the simulated annealing procedure used here), and having to find ways to deal with changes in the classification over the year. For example, a forecast issued in one season and ending in the next must deal with a jump in classification. There are methods for doing so (such as appending months to the traditional three-month seasons in the classification derivation), but these are again choices. Furthermore, the classification we used here, MO30, has strong seasonality despite being derived on an annual scale (Neal et al., 2016; Richardson et al., 2018), hence a decision not to apply the classification procedure to individual seasons.

In addition, we focussed on MO30 alone because its relationship with historical UK

drought has already been analysed (Richardson et al., 2018) and is used operationally by the UK Met Office and Environment Agency Flood Forecasting Centre for coastal flooding applications (Neal et al., 2018). Therefore, there is interest in exploring further applications using this classification specifically.

**Changes to manuscript:** We have added details of the frequencies of occurrence of each WP to Fig.1 (see the supplement) to highlight the strong seasonality exhibited by WPs in MO30. We have also added the following sentences in Section 2 highlighting the differences between the frequencies of the ERA-Interim WPs used here and the original WPs used to derive the classification and determine how the WPs were numbered (i.e. their ordering according to historical frequency):

"A consequence of assigning WPs using ERA-Interim compared to the EMULATE data set used in the original derivation of MO30 is that the historical frequencies of occurrence of the WPs differ. The same strongly seasonal behaviour is retained (lower-numbered WPs occurring more often in summer than higher-numbered WPs, and vice versa), but the annual frequencies are more evenly distributed across the WPs - there is no clear decrease in annual frequency as the WP number is increased."

**Detailed comments**

**L102:** *it is not clear if there is a post-processing of the reforecasts. Do you observe any drift with lead time? Is there a bias between the distribution of assigned WP for short and long lead time?*

**Response:** As mentioned in Section 2 L102-104, the only post-processing is the removal of a MSLP climatology from the MSLP reforecasts to generate the anomalies. This climatology is the same as that used in the derivation of the WP classification (Neal et al., 2016), which is a reanalysis data set (EMULATE MSLP). We did not remove a lead-time dependent bias from the forecast model as the distribution of assigned WPs for our chosen lead times does not change particularly, as we show in Figure 1 at the end of this document.

**Changes to manuscript:** None

**L111:** *same question for the forecasted precipitation. Is there a correction/post- processing applied?*

**Response:** We did not apply any post-processing to precipitation. In hindsight, this does not provide a fair assessment of the skill of ECMWF precipitation forecasts (EPS-P) due to systematic model bias, and the fact that we are comparing to WP-based models that sample from the same observational precipitation data set that is used as the 'truth' in the forecast verification. As we frame the paper around the potential usefulness of WP methods compared to model precipitation, we feel that it is important to apply some post-processing to EPS-P.

We have now applied a 3-monthly-mean bias correction to these forecasts and regenerated the skill scores. Unsurprisingly, this increases the skill of EPS-P overall, and weakens the advantages that EPS-WP had over EPS-P as described in the original manuscript.

We thank the referee for highlighting to us that a calibration of EPS-P would make for a much fairer model comparison.

**Changes to manuscript:** We have added in a sentence to Section 2 to clarify that we have applied a bias correction to the precipitation forecasts. The results section has significantly changed as a result of this, in particular the section describing the forecast accuracy (RPSS and BSS results. The ROC and reliability scores are less affected). The modifications to the results section of the manuscript are too long to put here, we ask the reader to refer to the supplement that contains the new results sections.

Furthermore, we have removed some text (L367-385) from the conclusions recommending models to forecasters as the skill is now too similar to provide a clear choice, and changed the sentence on L352-353 to read:

"We compared two levels of drought: mild drought, when the total precipitation over the lead-time (16, 31 or 46 days) was below the 30.9th percentile climatology, and moderate drought, when the total precipitation over the lead-time was below the 15.9th percentile. Overall, forecast models were found to be more skilful during winter and autumn, particular for longer lead-times. The Markov model tended to be the least skilful, especially when predicting drought. Differences in skill between EPS-P and EPS-WP were typically small, with RPSS, BSS and ROC results not highlighting a clear winner."

**4.1:** *I think the improvement of forecast with lead time deserves more attention. This result should be better analyzed. The classification of WP is done with reanalysis, correct? The sentences, "the observation and the forecasts tend towards climatology at longer lead time' and 'As the lead-time is increased, the observations become noisier ...' sound weirds to me. There is no 'lead time' for the observations. Please clarify.*

**Response:** The skill of EPS-WP does not change much with lead-time – the JSD remains near 0.2 for all months and lead-times. The skill of the Markov model, however, increases significantly with lead-time. This is a consequence of using the JSD, which is a way of measuring the distance between the probability distribution of the observed WP frequencies of occurrence and the probability distribution of the forecasted WP frequencies of occurrence. The number of WPs in our classification is 30 and therefore these two probability distributions consist of 30 points, each reflecting how often every WP has occurred in the forecast or corresponding observed period. It is important to bear in mind that the length of the observed period is the same length as the forecast, so for forecast of 16 days, the observed period is also 16 days, while for 46-day forecasts, we compare with observations over 46 days.

For shorter lead-times, the probability distribution of observed WPs will be far noisier than the corresponding distribution from the Markov-predicted WPs, as different reali-sations of the Markov model (i.e. its ensemble members) diverge from each other very quickly. For longer periods, more WPs are likely to have occurred, resulting in a less noisy observed distribution. This is what we mean by "noise" in the observations for longer lead-times. The apparent increase in skill with longer lead-times for Markov, then, is a result of the distance in probability between the two distributions decreasing because of a smoothing of the observed WP frequencies distribution. That is, Markov is better at predicting the long-term average than the short-term.

In summary, for EPS-WP the distribution of forecast WPs looks as similar to the distri-bution of the observed WPs for shorter and for longer leads. For Markov, the distribution of predicted WPs looks far more like the distribution of observed WPs at longer leads than at shorter leads.

The reason we did not use typical verification metrics here, such as a Brier Score mea-suring the models' ability to predict WPs at each lead independently (through hits and misses), is that we are considering the total precipitation over multiple weeks. There-fore, we are not interested particularly in whether the models correctly predict the timing of a WP, only that they capture the general distribution over each forecast period.

**Changes to manuscript:** We have modified the second paragraph of Section 4.1 to be clearer as to the above explanation. Following a comment by the other referee, we have added some text to highlight the fact that the JSD might not be a particularly useful verification metric in the traditional (operational) sense. The second and third paragraphs of Section 4.1 now read:

"An interesting result is how JSD scores for Markov decrease as the lead-time in-creases (Fig. 3), suggesting an improvement in skill with lead-time. This is the opposite of the expected (and usual) effect. The Markov model predicts WPs using the one-day transition probabilities, and its ensemble members therefore diverge very quickly, re-sulting in a distribution of predicted WPs that looks similar to the climatological WP

distribution for all lead-times. For a 16-day forecast, the observed WP distribution of the corresponding 16 days will generally be less similar to the climatological WP distribution than for 31-day forecasts, and less similar still than for 46-day forecasts. For instance, at a 16-day lead, only 16 unique WPs could form the observed distribution, whereas Markov is capable of predicting all possible WPs across its 1000 members at this lead. As the JSD measures the distance between these probability distributions, it tends to score the differences between these distributions as more similar (a smaller divergence) for longer lead-times. This means the JSD is perhaps not appropriate as a verification metric in an operational sense, but is noteworthy for highlighting the behaviour of the Markov model.

We could have assessed model skill in predicting the WPs using more common metrics such as the BS, which could measure the hit/miss ratio for each WP at each lead-time. However, the focus of this paper is on multi-week precipitation (and drought) totals, so we are not particularly interested in the models' ability to predict the timing of a WP, only whether they are able to capture the distribution of the WP frequencies of occurrence. It is likely that using the BS would show that EPS-WP and Markov skill decreases with lead-time, as was the case for a WP classification derived from MO30 by Neal et al. (2016)."

**L238:** *How do the authors explain the fact that EPS-WP outperforms Perfect-WP in summer? That could reflect a bias in the forecasted WP compared to the observed ones.*

**Response:** There are no cases where EPS-WP outperforms Perfect-WP in Fig. 4 (or elsewhere). Perhaps the referee has mistaken the EPS-P results for EPS-WP in this case?

**Changes to manuscript:** None.

**L250:** *"The difference in skill . . . for northern and western regions..." this will be more visible if the authors use maps instead of radar-plots.*

**Response:** This has been implemented as explained previously.

**L265:** *Maybe the sentences are a bit too optimistic. Indeed, for several regions, the differences between EPS-WP and EPS-P do not look significant (ES, NEE, SEE in winter for different lead times). The potential benefit of the use of WP (perfect-WP) could be also discussed since if the WP are not well forecasted and if the processes between WP and precipitation is well represented in the model, that means the main mistake of the model is the same when using forecasted WP and precipitation. That could limit the interest of using such predictors. This should be discussed.*

**Response:** We agree that these sentences are a little optimistic, particularly since we corrected the bias of EPS-P, reducing the difference in skill between this model and EPS-WP. We also agree with the referee regarding the limitations of WPs as predictors given the difficulty EPS-WP has in predicting them.

**Changes to manuscript:** As mentioned in a previous comment regarding precipitation post-processing, we have updated the results section (see supplement) and removed some recommendations to forecasters from the conclusions. We have also expanded on our discussion of potential improvements to increase the predictability of the WPs by adding the following sentences to the last paragraph of the conclusions section:

"The skill of this model during winter and autumn suggest that the processes between the WPs are precipitation are well represented in these seasons. The lesser skill of EPS-WP and Markov, then, is a result of poor prediction of the WPs. A focus on improving the skill of the WP forecasts could be the most useful route to improving precipitation and drought predicting skill."

**L266:** *Does "simple statistical WP prediction" mean Markov here? I suggest to keep*

[Figure]

*the same name of the experience.*

**Response:** Yes, we were referring to Markov.

**Changes to manuscript:** In the new results section (see supplement), this phrase is no longer used.

**L279 and Fig 6:** *Please remind to the readers that mild droughts mean here. Also I am a bit confused about the definition of droughts with lead time d-46. Droughts are calculated with 30-d cumulated periods. Since the authors used the Extended ensemble, how they calculated droughts with 16 and 46-d lead time? Please clarify.*

**Response:** Droughts are not calculated with 30-day cumulated periods. As stated in Section 3.4 L168-170, they are calculated on cumulated periods of length equal to the forecast lead-time i.e. over 16, 31 or 46 days. However, we noticed that Table 2 had the incorrect heading of "30-day precipitation sums".

**Changes to manuscript:** We have reminded readers of the definitions for mild and moderate drought at the beginning of Section 4.3.1 and in the figure captions, as suggested. We have also added clarification when originally defining the types of drought to stress that droughts are defined according to climatological percentiles derived separately for each lead. We have also modified Table 2 – both the incorrect heading and to add a row indicating the max precipitation intervals, as suggested in RC2. See the supplement.

**L284 (and elsewhere):** *Because of the radar-plots it is quite complicated to locate the regions (here Scotland). That requires constantly back and forth to the UK map (Fig. 2).*

**Response:** See earlier regarding new map plots.

**L292:** *It is not clear why the authors conclude "... but not for more severe droughts."*
*since these results are not shown nor discussed previously. Does it mean the same*
*results with severe droughts provide negative BSS? These results could be discussed*
*and added in Sup Mat.*

**Response:** Thank you for pointing out this error. We were referring to moderate
drought, not to a more severe class.

**Changes to manuscript:** This phrase is no longer in the results section since it has
been updated (see supplement).

**L313:** *"EPS-WP is the most reliable forecast model" The authors should clarified that*
*perfect-WP is the most reliable but not a real forecast.*

**Response:** Ok, thanks for the suggestion.

**Changes to manuscript:** The relevant sentence now reads:

"EPS-WP is the most reliable forecast model (i.e. excluding Perfect-WP). . ."

**L325:** *"... forecasts from EPS-WP are more reliable than from Perfect-WP. . ." Accord-*
*ing to several figures, I am not convinced with that conclusions (Fig. 9c, 9g, 10a, 10c,*
*10g)*

**Response:** We drew this conclusion based on the fact that the results for EPS-WP lie
closer to the diagonal than for Perfect-WP. We think that this conclusion is fair, although
only for drought forecasts issued with a probability below a certain level (80% for mild
drought and 60% for moderate drought). We think this is an important clarification to
add to the paper.

**Changes to manuscript:** We have added to the sentence on L325-326, which now
reads:

"An interesting result is that forecasts from EPS-WP are more reliable than from Perfect-WP when the predicted drought probabilities are below 80% for mild drought (Fig. 9) and 60% for moderate drought (except in spring; Fig. 10), despite having lower accuracy (e.g. Fig. 6)"

And a sentence on L328:

"However, for drought forecasts issued with higher probabilities, EPS-WP is the less reliable model, under- or over-forecasting drought (depending on the season) more than Perfect-WP."

**Discussion section:** *In the recommendation section, the authors should redefine their drought definition. Also these conclusions could be different if they change the definition of WP (by splitting the year in 2 or 4 seasons).*

**Response:** We think it is a good idea to redefine our drought definitions here. We agree that the conclusions could be different if the WP classification was derived separately for each season. They could also be different if we made other choices when defining the classification e.g. domain size, number of WPs etc. See our response to a previous comment (page C3 of this document).

**Changes to manuscript:** We have redefined drought definitions in this section, as suggested.

[Figure]

**Fig. 1.** Frequency of forecast WP by the ECMWF EPS at lead times of 16, 31 and 46 days.

**Supplement:**

Supplement for RC1 and RC2 responses

This supplement contains the new text relating to Sections 4.2 and 4.3.1 of the original
submitted manuscript, which have changed as a result of applying a bias-correction to
ECMWF EPS precipitation and hence drought forecasts (EPS-P).

We also include all new plots e.g. a new model schematic and updates of the radar plots as
map plots. All figures are after the text.

[revised manuscript text omitted]

---

## Author Comment (AC2) · 23 Oct 2019

[revised manuscript text omitted]